# Optimization and Simulation of Mountain City Land Use Based on MOP-PLUS Model: A Case Study of Caijia Cluster, Chongqing

**Yuqing Zhong** [1,2,3]**, Xiaoxiang Zhang** [1,2,3,*] 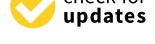**, Yanfei Yang** [1,2,3] **and Minghui Xue** [1,2,3]

1   College of Geography and Remote Sensing, Hohai University, Nanjing 211100, China;
    211301060014@hhu.edu.cn (Y.Z.); yanfei.yang@hhu.edu.cn (Y.Y.); xueminghui@hhu.edu.cn (M.X.)
2   Institute of Geographic Information Science and Engineering, Hohai University, Nanjing 211100, China
3   Center for Geospatial Intelligence and Watershed Science, Hohai University, Nanjing 211100, China
*   Correspondence: xiaoxiang@hhu.edu.cn

**Abstract:** Mountainous cities face various land use challenges, including complex topography, low land use efficiency, and the insufficient control of land use in small-scale areas at the urban fringe. Considering population changes, environmental conservation, and urban planning, this study first established three scenarios: economic priority (Econ. Prior.), ecological priority (Ecol. Prior.), and balanced development (BD), and then used the Multi-Objective Planning (MOP) model to calculate the optimal land use structure. Finally, it carried out land use spatial layout optimization based on the Patch-generating Land Use Simulation (PLUS) model in 2035, Caijia Cluster, Chongqing, China. This approach, known as MOP-PLUS modeling, aimed to optimize land use. Meanwhile, the applicability of the PLUS model in simulating land use changes was discussed in small-scale mountainous areas. The results show the following: (1) The "quantity + space" approach in the MOP-PLUS model demonstrated the feasibility of the PLUS model in simulating land use change in small-scale mountainous areas. The overall accuracy (OA) of land use change simulation reached 81.60%, with a Kappa value of 0.73 and a Figure of Merit (FoM) coefficient of 0.263. (2) Land use optimization: Under the Econ. Prior. scenario, economic benefits peaked at $4.06 \times 10^{10}$ CNY. Urban expansion was the largest, leading to increased patch fragmentation. The Ecol. Prior. scenario yielded the highest ecological benefits, reaching $7.46 \times 10^7$ CNY. The urban development pattern exhibited inward contraction, accompanied by urban retrogression. In the BD scenario, economic benefits totaled $3.89 \times 10^{10}$ CNY, and ecological benefits amounted to $7.16 \times 10^7$ CNY. Construction land tended to concentrate spatially, leading to relatively optimal land use efficiency. Therefore, based on a comprehensive consideration of the regional land use constraint policies and spatial layout, we believe that a balance point for land use demands can be found in the BD scenario. It can ensure economic growth without compromising the ecological environment.

**Keywords:** Caijia Cluster; construction land; land use structure optimization; spatial layout optimization; MOP-PLUS model

## 1. Introduction

As urbanization accelerates and the population migrates to urban areas, the urban boundaries expand continuously, resulting in corresponding changes in the land use structure of the urban fringe. This transformation is characterized by the expansion of urban infrastructure, commercial districts, and residential areas, often at the cost of cultivated land and forests [1]. This trade-off aims to maintain a balance in the total land area within the region, following a "one gain, one loss" principle. In essence, urbanization involves the reconfiguration of various land resources. High-speed urbanization brings about significant alterations in land cover. (1) The rapid expansion of impervious surfaces like construction land contributes to the urban heat island effect, which has a substantial impact on regional

climate change and accelerates climate warming [2,3]. (2) Population migration and the expansion of construction land reshape the terrestrial ecological landscape, leading to various ecological and environmental challenges, including changes in the bird biodiversity [4] and the destruction and restoration of the surface vegetation [5–7]. Urbanization is an essential pathway for societal progress and development. However, it is crucial to strike a balance between construction land expansion and non-construction land preservation. This balance is necessary to address the inherent contradiction between socio-economic development and environmental conservation, ultimately facilitating the rational allocation of regional land resources.

Currently, researchers in the field of land resource allocation mainly pay attention to land use quantity structure and spatial layout optimization. The former focuses on optimizing and regulating the allocation of territorial space [8]. It concerns land ecological benefits, such as ecosystem service value [9,10] and global climate change considerations [11]. For instance, Searchinger et al. [12] proposed a carbon benefits index to assess the impact of land use change on global carbon emissions. As for the latter [13], many scholars engage in exploratory research related to urban space. This exploration includes spatial modeling, algorithm optimization, and geographical information system (GIS) analysis [14]. As an example, Han et al. [15] gathered mobile phone signaling data and point of interest (POI) data to analyze the geographical distribution characteristics of Beijing's urban industry and commerce (accommodation, catering, etc.). In the context of territorial spatial planning, an increasing number of researchers have studied the optimization methods for land resources allocation by considering the indicator systems of land use suitability [16,17] and resource environmental carrying capacity evaluation [18]. Substantial works lead to significant contributions in domains such as combining territorial space models [19], defining the division of "production-living-ecological space" [20], and optimizing spatial layouts [21].

Regarding research methods, both domestic and international scholars primarily adopt a combination of land use structure and spatial layout optimization methods. They conduct research by integrating multiple models to control changes in the quantities of land use types and to perform spatial layout optimization [22–24]. Specifically, Multi-Objective Planning (MOP) [25] is used to regulate changes in land use quantities, while various land use/land cover (LULC) simulation models such as the Conversion of Land Use and its Effects at Small regional extent (CLUE-S) [26–28], Cellular Automata (CA) [29,30], Future Land Use Simulation (FLUS) [31,32], advanced FLUS [24], and Patch-generating Land Use Simulation (PLUS) [33,34] are applied for spatial layout optimization. Furthermore, in terms of research scales, many scholars select different study area scales for simulating land use changes. At larger scales, such as basins [35], urban agglomeration [36], and metropolitan areas [37], the objective is to explore the ecological issues and propose optimization solutions related to urban land use changes. For instance, Lou et al. [38] studied the impact of land use change scenarios on the ecosystem service value of the Yellow River Basin, and Salazar et al. [39] conducted the land use changes in the Quito metropolitan area and its related urban agglomerations by combining risk assessment and nature conservation. Regarding smaller scales, like county-level urban land use analysis [39–41], the focus is on studying land use suitability, efficiency, and their influencing factors at the county level. As for even smaller scales, like townships [42], the analysis is focused on land use evolution characteristics within townships. Currently, researchers choose economically developed cities or regions in the eastern part of the country to simulate land use expansion. However, there is limited research on land use changes in highland mountainous areas with significant topographical variations during urbanization.

The challenging task of balancing socio-economic development and environmental conservation is amplified by the complex terrain and limited land resources in mountainous cities [43]. Land use issues are significant in small-scale areas at the urban periphery. Here, land management authorities often lack the necessary constraints and regulations over changes in rural and township land use. Villagers frequently occupy cropland for various construction purposes, such as housing and roads, due to the absence of detailed land

management systems. In regions where land use issues are more pressing, addressing the specific land use needs and characteristics through detailed solutions becomes imperative. There is an urgent need to optimize the allocation of land resources in small-scale areas and make timely adjustments to land use structure and direction. The PLUS model can accurately simulate the relationships behind land use type conversions [44]. It enables the simulation of land use layouts in complex terrain and reflects their dynamic changes. The PLUS model incorporates two land use scenario simulations: linear regression and Markov-chain scenarios. However, it lacks sensitivity to urban planning policies, population mobility, and macroeconomic policies, and its simulated future land use changes lack sufficient empirical basis. Some researchers primarily restrict land use quantities in the MOP model based on regional land use planning policies [24,33], without considering the constraints of social factors such as population and environmental changes. Hence, this study, based on land use planning policies, integrates population and ecological environmental changes as constraints within the MOP model. This comprehensive (MOP-PLUS model) approach fully considers local land use demands and environmental pressures. It optimizes land type allocation in small-scale areas, balancing the demands of various land use types, including cropland, construction land, and so on. Furthermore, it compares the spatial distribution of various land use type changes under different development scenarios using the PLUS model. This paper aims to study the applicability of the MOP-PLUS models in optimizing land use simulations in small-scale mountainous areas and provide scientific evidence and reference for the standardization of future land use in small-scale areas and the formulation of land use planning policies.

## 2. Study Area and Data Used

### 2.1. Study Area

The study area in Figure 1 is located in the north of Chongqing's central metropolitan area and the south of Beibei District. It occupies a strategic location, serving as a pivotal link between Beibei District, Jiangbei District, and Shapingba District. The area plays a crucial role as a gathering place for population and industries, as well as a driving force for the economic development of Beibei District and even the central metropolitan area. The Cluster is backed by Zhongliang Mountain and is surrounded on three sides by the Jialing River. The study area includes Caijiagang Subdistrict, Shijialiang Town, and Tongjiaxi Town, which covers an area of 88 km$^2$. Currently, approximately 55 km$^2$ of land within the designated construction area is available for development, with a planned population of 610,000. The area has been designated as an inland open economic pilot zone in Chongqing, entitling it to preferential policies associated with national-level development zones. Supported by a variety of urban development policies, the study area has entered a new phase of urban growth, capitalizing on its strategic location, natural environment, transportation networks, and other advantages.

### 2.2. Data Sources and Preprocessing

The data and materials required in this study can be divided into four parts: socio-economic data, land use data, fundamental geographic data, and other data. The data sources and explanations are listed in Table 1.

(1) Socio-economic data. The unit economic benefits of forestry, husbandry, and other output values used to calculate the economic benefit coefficients were extracted from the "Chongqing Statistical Yearbook" in 2015–2018 (Table 2). Population density and Gross Domestic Product (GDP) grid data in 2015 were sourced from the Resource and Environment Science and Data Center, Chinese Academy of Sciences, which were used as socio-economic drivers of land use changes. Relevant planning materials referenced included "Chongqing Land Use Master Planning (2006–2020)" (referred to as "Land Use Planning"), "Territorial Spatial Planning of Chongqing Municipality (2021–2035)" (referred to as "Spatial Planning"), and "Current Land Use Classification (GB/T21010-2017)" [45] (referred to as "Land Use Classification").

(2) Land use data. Landsat-8 images with a resolution of 30 m were obtained from the Google Earth Engine (GEE) platform (https://code.earthengine.google.com (accessed on 6 June 2019)). The image with the least contemporaneous cloud cover was selected for data processing (e.g., clipping, cloud removing) to enhance classification accuracy. The Landsat image acquisition dates are provided in Table 1. The Random Forest (RF) and supervised classification methods were employed to process the data. Based on the local land use characteristics and the standard "Land Use Classification", the images were classified into seven land use types, including cropland, forest, water, bare land, impervious surface, shrubland, and grassland. The interpretation results met the required accuracy. The land use data served as input for the PLUS model, which was used for accuracy validation and simulating land use changes.

(3) Fundamental geographic data. This category includes administrative divisions, the Digital Elevation Model (DEM), transportation networks, railway stations, and soil-type data. The DEM was processed using slope analysis in ArcGIS. Buffer analysis was performed to calculate the distances from the study area to main roads, various levels of roads, highways, and train stations. Except for administrative divisions, all other fundamental geographic data were utilized as driving factors in the PLUS model.

(4) Other data. Net Primary Productivity (NPP) and Normalized Difference Vegetation Index (NDVI) at a 1 km resolution were incorporated to adjust the ecological benefit coefficient in the MOP model. The obtained data were uniformly projected in the WGS-1984 coordinate system and resampled to a spatial resolution of 30 m × 30 m.

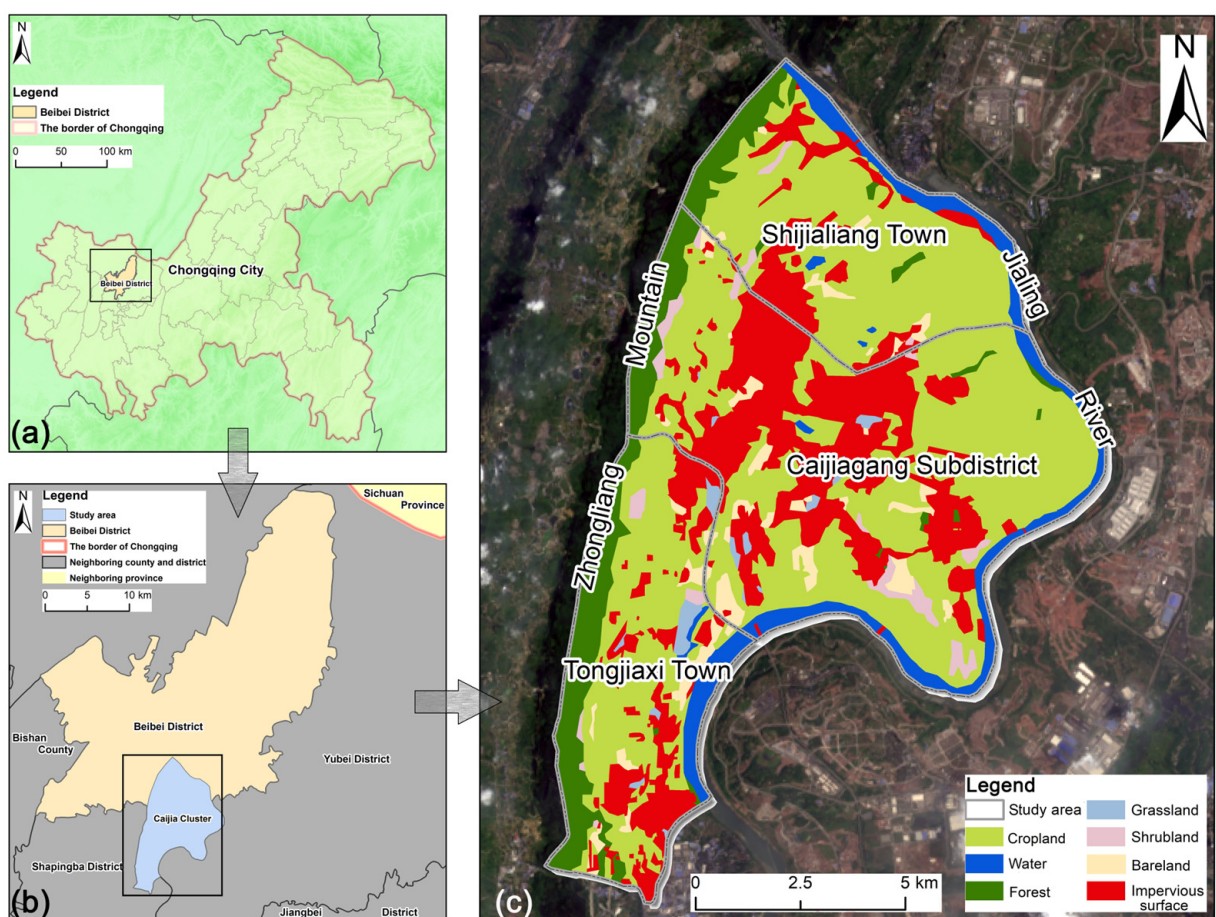

**Figure 1.** Study area: (**a**) The location of Beibei District, (**b**) The location of Caijia Cluster, (**c**) The three administrative districts within the study and land use status in 2018.

**Table 1.** The list of collected data and reference material in this study.

| Category | Data/Material | Resolution | Source | Data Usage |
|---|---|---|---|---|
| Socio-economic data | Economic statistical data | | Chongqing Statistics Bureau | Calculation of economic efficiency factor |
| | Density of population GDP | 1 km 1 km | Resource and Environment Science and Data Center | Socio-economic drivers |
| | Chongqing Land Use Master Planning (2006–2020) Territorial Spatial Planning of Chongqing Municipality (2021–2035) Current Land Use Classification (GB/T21010-2017) | | Chongqing Planning and Natural Resources Bureau | Reference material |
| Land use/cover data | Landsat 8 | LC08_L2SP_128039_20150708 LC08_L2SP_128039_20160710 LC08_L2SP_128039_20170915 LC08_L2SP_128039_20180902 | GEE | PLUS model input |
| | | 30 m | | |
| Fundamental geographic data | DEM | 30 m | Geospatial Data Cloud | Natural factor drivers |
| | Soil types | 1 km | Resource and Environment Science and Data Center | |
| | Transportation networks Railway stations Administrative division | | Open Street Map Chongqing Geographic Information Center | Traffic accessibility drivers Relevant analysis |
| Other data | NPP NDVI | 1 km 1 km | Resource and Environment Science and Data Center | Modification of the ecological efficiency factor |

**Table 2.** The gross output value of the economic sectors in 2015–2018 (unit: $10^4$ CNY).

| Year \ Economic Sectors | The Secondary and Tertiary Sectors | Forestry | Farming | Husbandry | Tea Gardens and Orchards | Fishery |
|---|---|---|---|---|---|---|
| 2015 | 2,417,527.74 | 573.76 | 12,560.73 | 8846.65 | 774.47 | 4695.26 |
| 2016 | 2,680,109.26 | 676.52 | 14,493.15 | 7465.13 | 906.15 | 5185.99 |
| 2017 | 2,868,781.64 | 1010.50 | 14,309.11 | 7324.87 | 1204.15 | 6985.34 |
| 2018 | 2,866,303.04 | 1224.59 | 15,551.38 | 7481.78 | 1325.44 | 7324.61 |

CNY = Chinese Yuan.

## 3. Methodology

The methodology in this study can be divided into two main parts: land use structure optimization and land use spatial layout optimization.

(1) The optimization of land use structure was based on the MOP model, which involved the establishment of urban development scenarios that prioritized economic development, ecological sustainability, and a balance of them. Subsequently, the Grey Forecasting (GF) model was utilized to predict economic benefit coefficients and the equivalent factor method was employed to adjust ecological benefit coefficients. This allowed for multi-scenario optimization of land use structure within the study area while adhering to the constraints set by the MOP model.

(2) The optimization of spatial layout, based on the PLUS model, considered three categories of factors: natural, societal, and transportation related. The RF algorithm was applied to determine the development probabilities of various land use types. Using the land use quantity structure optimized by the MOP model as input and incorporating the development probabilities of different land use types as rules, the spatial layout of land use was optimized under three development scenarios. The overall framework is illustrated in Figure 2.

### 3.1. MOP Model

The MOP transforms inherently incomparable multiple objectives into a single objective optimization problem. Multi-objective linear planning consists of two or more objective functions and multiple constraints, to obtain optimal values or maximize benefits

for multiple objectives [46]. The MOP model is a vital mathematical tool employed in land use optimization research, geographic studies, and land resource management.

**Figure 2.** The overall research framework: (**a**) Datasets used; (**b**) The method of MOP-PLUS model; (**c**) Multi-scenario land use structure and spatial layout optimization results analysis.

The formulation is as follows:

$$f_1(x) = max \sum_{i=1}^{n} d_i x_i \qquad (1)$$

$$f_2(x) = max \sum_{i=1}^{n} e_i x_i \qquad (2)$$

$$s.t. \sum_{i=1}^{n} C x_i = (\leq, \geq) \mathrm{B} \qquad (3)$$

where $f_1(x)$ and $f_2(x)$ represent the objective functions for economic benefits and ecological benefits. $d_i$ and $e_i$ denote the unit area economic and ecological benefit coefficients of different land use types. $x_i$ is the decision variable for the *i*-th land use type. *s.t.* is the constraints, *C* is the constraint coefficients, and B represents voluntary restrictions.

### 3.1.1. Development Scenario Setting

To achieve regional socio-economic goals while ensuring ecological stability, it is necessary to protect cropland and the ecological environment, scientifically allocate construction land, and optimize land use structure and spatial layout. In this study, based on the advantages of development, resource endowment, ecological conservation, and urban planning in the study area, three development scenarios were set as follows:

(1) Economic priority (Econ. Prior.) scenario: This scenario prioritized the utilization of land resources with higher economic benefits to improve the efficiency of land use, promote urban–rural integration, and meet the needs of economic development [47]. Land development and construction were primarily focused on land types with higher economic benefits, such as construction land, industrial areas, and water.

(2) Ecological priority (Ecol. Prior.) scenario: Due to the non-renewability of natural resources and the necessity of ecological civilization construction, this scenario emphasized

maintaining the existing ecological forest and grassland, implementing land restoration and afforestation measures, and preserving the ecological balance to maximize ecological benefits.

(3) Balanced development (BD) scenario: Focusing solely on economic development or ecological protection goals was not conducive to the healthy and sustainable development of the region. To realize a more harmonious land use structure and spatial layout, this scenario balanced ecological and economic objectives, aiming to obtain a set of non-dominated optimal solutions.

### 3.1.2. Decision Variables and Constraints

Based on the results of land use classification for the study area in Section 2.2, seven decision variables were established: impervious surface ($x_1$), forest ($x_2$), cropland ($x_3$), grassland ($x_4$), shrubland ($x_5$), water ($x_6$), and bare land ($x_7$).

Land use optimization needs to consider multiple factors to achieve a reasonable quantity structure and improve land use efficiency. When setting constraints, it is essential to align with local development planning requirements, ensure macroeconomic regulation, and maintain the relationship with the current land use structure. Based on the ecological environment, the socio-economic conditions, the land use status in 2018, and the preliminary requirements and proposals from the "Land Use Planning" and the "Spatial Planning", seven constraints were established, including total area, planning population in 2035, etc. The constraints are shown in Table 3.

### 3.1.3. Objective Function Construction

(1) Construction of the Economic Benefit Objective Function. According to Table 2, the ratio of the output value in the secondary and tertiary sectors was used as the economic benefit coefficient for construction land. The economic benefit coefficients for forest and cropland were determined based on the output value per unit area of forestry and crops (excluding tea gardens and orchards). The economic benefit coefficients for shrubland and grassland were based on the output value per unit area of tea gardens and livestock industry, respectively. The economic benefit coefficient for water was calculated using the output value of fisheries and aquaculture. The economic benefit coefficient for bare land was set as 0.0001 based on the literature references [48]. The function was constructed as follows:

$$f_1(x) = 1538.75x_1 + 4.61x_2 + 6.16x_3 + 86.41x_4 + 234.98x_5 + 162.69x_6 + 0.0001x_7 \quad (4)$$

where $x_1$ to $x_7$ represent the areas of construction land, forest, cropland, grassland, shrubland, water, and bare land, respectively.

(2) Construction of the Ecological Benefit Objective Function. Considering data availability and method feasibility, the equivalent factor method was used to calculate the ecological value of ecosystem services. The Chinese Land Ecosystem Service Value Equivalent Table proposed by Xie [49] and the Chinese Land Ecosystem Service Value Table for different land ecosystems were referenced. The ecological value coefficients were adjusted using the value coefficient adjustment method proposed by Pan [50] to improve the accuracy of the per unit area ecological service value. The objective function was constructed as follows:

$$f_2(x) = 0.0001x_1 + 1.94x_2 + 0.61x_3 + 0.67x_4 + 1.27x_5 + 4.07x_6 + 0.04x_7 \quad (5)$$

where $x_1$ to $x_7$ represent the areas of construction land, forest, cropland, grassland, shrubland, water, and bare land, respectively.

**Table 3.** The constraint conditions of MOP in this study.

| Constraint | Formula | Description |
|---|---|---|
| Total area | $\sum_{i=1}^{7} x_i = S$, $S = 8770.69$ (ha) | The total area of the study area should remain unchanged. |
| Planning population in 2035 | $a_1 \times x_1 + a_2 \times (x_2 + x_3 + x_4 + x_5) \leq b_1$ | Based on the urban and farmland population density and total population in Chongqing from 2015 to 2018, the values in 2035 are projected using the GF model as follows: $a_1 = 75.36$, $a_2 = 4.36$ (persons per hectare), $b_1 = 487{,}787$. By 2035, the total population is not to be larger than 487,787. |
| Ecological environment | $x_2 + x_3 + x_4 + x_5 \geq 5718.28 \times \theta$ | With ecological environment protection as a constraint, it is ensured that the areas of forest, cropland, grassland, and shrubland are within an appropriate range $\theta$. The value 5718.28 represents the total area of $x_2$, $x_3$, $x_4$, and $x_5$ in 2018. |
| Flexible planning | $0.75 \leq \theta \leq 1.25$ | The elasticity coefficient $\theta$ is set within the interval [0.75, 1.25]. It was determined based on the proportions of cropland, forest, grassland, and shrubland as specified in the "Land Use Planning", taking into account historical trends and the length of the forecasting period. |
| Policies | $2136.72 \leq x_1 \leq 2513.63$, $x_5 \geq 173.26$ | According to "Spatial Planning", the urban and rural construction land is projected to increase by 17.64% in 2035, with a stable increase in the area of shrubland. Therefore, the area of construction land should be between 2136.72 and 2513.63 (ha), with the shrubland area not being less than 173.26 (ha). |
| Land development utilization rate | $\frac{S - x_7}{S} \geq r$, $x_7 \leq 309.85$ | The land development utilization rate ($r$) in 2035 should not be lower than that in 2018; the bare land area should be greater than 309.85 (ha). $r_{2018}$ is calculated using the following formula: $r_{2018} = ((S - x_{7(2018)})/S) \times 100\%$. |
| Decision variable non-negative | $x_i \geq 0 (i = 1, 2, ..., 7)$ | The area of each land use type should not be less than 0. |

*3.2. PLUS Model*

3.2.1. Overview of the PLUS Model

The PLUS model is based on traditional CA and incorporates the improved RF algorithm, adaptive inertia, and competition mechanisms. It provides higher precision and is suitable for simulating multiple land use patches and optimizing land use patterns under different development scenarios. The PLUS model framework mainly includes the Land Expansion Analysis Strategy (LEAS) for transformation rule mining based on land expansion analysis and the CA model based on multi-type Random Seeds (CARS) using a multi-class random patch seed mechanism.

(1) LEAS module. The LEAS module conducts overlay analyses of two land use data sets, identifies locations with changes between the two periods, and randomly selects sample points. The RF algorithm is used to train different land use data and obtain transformation rules for land use expansion.

The RF algorithm is used to randomly sample various land classes to output the growth probability $P_{i,k}^d$ of land use type k at grid *i*. The formula can be expressed as follows:

$$p_{i,k}^d(x) = \frac{\sum_{n=1}^{m} I(h_n(x) = d)}{m} \tag{6}$$

where $d$ can only be 1 or 0, with $d = 1$ indicating a conversion from other land use types to type k, and $d = 0$ otherwise. $x$ represents the driver factor vector, $I(\cdot)$ is a function related to decision trees in the RF, $h_n(x)$ is the predicted type of vector x in the $n$-th decision tree, and $m$ is the number of all decision trees.

The LEAS module obtains the growth patches of changing land classes without considering their sources, simplifying the analysis of land use changes. The transformation rules obtained from LEAS have a time attribute and can describe the characteristics of land use change within a specific time interval.

(2) CARS module. The PLUS model combines the development probabilities of various land use types, neighborhood weights, conversion rules, and regional constraints to achieve a rational allocation of land use types in pixel space. To simulate the evolution of multiple land use types, the PLUS model adopts a threshold-based multi-type random seed patch mechanism. This mechanism uses the statistical sampling method (Monte Carlo method) to calculate the development probabilities of land use types when the neighborhood effect of a certain land type is 0, generating "seeds" of change based on the output of LEAS. The seed generation rules are as follows:

$$OP_{i,k}^{1,t} = \begin{cases} P_{i,k}^1 \times (r \times \mu_k) \times D_k^t \ if \ \Omega_{i,k}^t = 0 \ and \ r < P_{i,k}^1 \\ \qquad P_{i,k}^1 \times \Omega_{i,k}^t \times D_k^t \quad all \ others \end{cases} \tag{7}$$

$$\Omega_{i,k}^t = \frac{con\left(c_i^{t-1} = k\right)}{n \times n - 1} \times w_k \tag{8}$$

$$D_k^t = \begin{cases} D_k^{t-1}, \ if \ \left|G_k^{t-1}\right| \leq \left|G_k^{t-2}\right| \\ D_k^{t-1} \times \frac{G_k^{t-1}}{G_k^{t-2}}, \ if \ \left|G_k^{t-1}\right| > \left|G_k^{t-2}\right| > 0 \\ D_k^{t-1} \times \frac{G_k^{t-1}}{G_k^{t-2}}, \ if \ G_k^{t-1} < G_k^{t-2} < 0 \end{cases} \tag{9}$$

where $OP_{i,k}^{1,t}$ represents the overall probability of land use type $k$ at grid $i$ and time $t$; $\Omega_{i,k}^t$ is the probability of land use type $k$ occurring in grid $i$, representing neighborhood influence; $D_k^t$ is the inertia coefficient of land use type $k$ at time $t$, which depends on the difference between the current quantity and the target demand; $r$ is a random number between 0 and 1; $\mu_k$ is the expansion threshold; and $w_k$ is the weight parameter for different land use types, varying with different neighborhood influences.

If a new land use type emerges as the selected choice in a round, the candidate land use types are chosen through roulette wheel selection based on the threshold decrease evaluation. The threshold decrease rule is as follows:

$$if \sum_{k=1}^{N} \left|G_c^{t-1}\right| - \sum_{k=1}^{N} \left|G_c^t\right| < Step \ Then, l = l + 1 \tag{10}$$

$$\begin{cases} Change \ P_{i,c}^1 > \tau \ and \ TM_{k,c} = 1 \\ Unchange \ P_{i,c}^1 \leq \tau \ or \ TM_{k,c} = 0 \end{cases} \tau = \delta^l \times r^* \tag{11}$$

where $Step$ is the step size required for simulating land use, $l$ is the decay step, $G_c^t$ and $G_c^{t-1}$ represent the differences between land use type $k$ in the current situation and future demand; $TM_{k,c}$ is the conversion matrix from land use type $k$ to $c$; $c$ is the new land use type; $\delta$ is the decay factor; and $r^*$ is a random value satisfying the normal distribution on [0, 2]. The adjustment of the parameters, $Step$, $l$, and $r^*$, is employed to control the simulation process and outcomes, adapting to different land development scenarios. In this phase, the focus primarily lies in configuring the land transfer matrix $TM_{k,c}$, which is largely determined by the optimization results of the multi-objective model to assess the transition probabilities of various land classes.

### 3.2.2. Rules and Parameter Settings

Considering the elevation and road characteristics of the study area, multiple factors including a multi-level road network, elevation, and slope were incorporated into the simulation to enhance its accuracy. This study, by reviewing the relevant literature and taking into account the specific conditions of the study area, selected 12 driving factors based on principles such as data availability and consistency, the quantifiability of the driving factors, and spatial heterogeneity, encompassing aspects related to the natural environment, socio-economic factors, and transportation accessibility. The drivers are shown in Table 4. The driving factors were used in the LEAS module to calculate with the RF. In the CARS module, the conversion land class layer was superimposed and, based on the historical land use change trend and different development scenarios, a land class conversion matrix was determined to determine the level of land expansion. Neighborhood weights were calculated by the proportion of each land use type's expansion area.

**Table 4.** The drivers in this study.

| Category | Driver |
| --- | --- |
| Natural factor | Elevation |
| | Slope |
| | Soil Types |
| Social factor | GDP |
| | Population density |
| Traffic accessibility factor | Distance to Train Station |
| | Distance to Highway |
| | Distance to Railway |
| | Distance to Main Road |
| | Distance to Primary Road |
| | Distance to Secondary Road |
| | Distance to Tertiary Road |

The parameter settings for the RF in the LEAS module were as follows: the number of decision trees was set to 50; the number of features for training the RF (mTry) was set to 12; and the sampling rate was set to 0.01 by default. In the CARS module, the neighborhood size was set to 3 by default; the decay coefficient for the threshold decrease was set to 0.9. A higher threshold indicates a lower probability of changes in patches with overall probability. The expansion coefficient, which measures the model's ability to generate new patches, was set to 0.1 by default. The neighborhood weight was filled based on the proportion of different land use types under different development scenarios. These parameter settings were referenced from the PLUS model manual [44].

### 3.2.3. Accuracy Validation

The PLUS model requires an accuracy validation of previous land use to ensure the rationality of future land use patterns. To determine the rationality of the land use simulation results, this study conducted accuracy validation from two dimensions: quantity accuracy and spatial accuracy.

(1) Quantity Accuracy. A comparison was made between the simulated land use results and the actual land use records in 2018.

(2) Spatial Accuracy. The accuracy of the PLUS model is mainly measured using the Kappa and the Figure of Merit (FoM) index by comparing the simulated results with actual land use data. The Kappa is a commonly used method to assess the spatial accuracy of raster land use types. It combines two parameters: user accuracy and map accuracy. The formula is as follows:

$$Kappa = \frac{p_s - p_r}{1 - p_r} \tag{12}$$

where $p_s$ is the proportion of land use types that are consistent between the actual and simulated results, representing the accuracy of the simulation, and $p_r$ is the expected accuracy under random conditions. The Kappa ranges from 0 to 1, where a value closer to 1 indicates better consistency and closer similarity between the simulated land use and the actual one.

The FoM index is mainly used to measure the consistency between the observed conversions in reality and the predicted conversions in the simulation. The specific calculation formula is as follows:

$$FoM = \frac{N_B}{N_A + N_B + N_C + N_D} \tag{13}$$

where $N_A$ represents the number of pixels that had actual conversions but did not convert in the simulation, $N_B$ represents the number of pixels that had actual conversions and were correctly simulated, $N_C$ represents the number of pixels that had conversions both in reality and in the simulation but with incorrect conversion types, and $N_D$ represents the number of pixels that did not have conversions in reality but converted in the simulation.

## 4. Results

### 4.1. Accuracy Validation Results

The ratio of correctly simulated pixels to actual pixels was calculated for each land use type. The results are listed in Table 5. The overall accuracy (OA) for the land use simulation in the study area in 2018 was found to be 81.60%, indicating relatively high accuracy. The simulation accuracy for bare land and construction land showed relatively lower performance, but the accuracy for other land use types was high. This suggests that bare land is prone to conversion to other land use types, mainly reflected in the simulation where the land use type is bare land while the current land use type is construction land. Based on the 2015 land use classification basis, this study conducted accuracy validation by comparing the simulated land use grid for 2018 in the study area with the actual land use grid for 2018 using the PLUS model. The obtained Kappa was 0.73, and the FoM was 0.263.

**Table 5.** Comparison between the Actual Pixel and Simulation Results of the study area in 2018.

| Land Type | Total Pixel | Correctly Simulated Pixels | Accuracy (%) |
|---|---|---|---|
| Impervious surface | 20,648 | 15,863 | 76.83 |
| Forest | 12,694 | 10,161 | 80.05 |
| Cropland | 44,360 | 35,204 | 79.36 |
| Grassland | 999 | 887 | 88.79 |
| Shrubland | 1627 | 1346 | 82.73 |
| Water | 14,457 | 13,507 | 93.43 |
| Bare land | 2987 | 2092 | 70.04 |

### 4.2. Land Use Structure Optimization Results

In this study, three urban expansion scenarios were established: Econ. Prior. scenario, Ecol. Prior. scenario, and BD scenario. The land use optimization structures for these three scenarios were calculated using Matlab 2022, considering constraints such as total area and population within the research area. The optimization results are presented in Table 6.

#### 4.2.1. Economic Priority Scenario

The study area generated economic benefits of $4.06 \times 10^{10}$ CNY and ecological benefits of $6.91 \times 10^7$ CNY. Compared with the status in 2018, the economic benefits increased by $5.90 \times 10^9$ CNY, while the latter decreased by $3.22 \times 10^6$ CNY. From the perspective of land use structure, the impervious surface increased by 376.91 ha in 2035. The generation of economic benefits mainly came from the growth of the secondary and tertiary industries, closely related to industries such as manufacturing, construction, and services. As a result, the area of impervious surface experienced significant growth to maximize economic benefits. Due to the relatively low economic benefit coefficients for forest, cropland, and bare

land, the optimized results showed obvious reductions compared with the 2018 baseline, with decreases of 170.65 ha, 26.12 ha, and 224.53 ha, respectively. However, cause the high economic benefit of shrubland, it experienced a slight increase compared with the ecological benefit, with an expansion of 51.10 ha. Overall, urban expansion was significant, resulting in a larger scale and greater encroachment on the forest, cropland, and bare land, causing a substantial loss of ecological benefits.

**Table 6.** Results of land use optimization structure under different development scenarios.

| Variable Symbol | Variable Name | Status in 2018 | Different Development Scenarios in 2035 | | | | | |
| --- | --- | --- | --- | --- | --- | --- | --- | --- |
| | | | Econ. Prior. Scenario | | Ecol. Prior. Scenario | | BD Scenario | |
| | | Area (ha) | Area (ha) | Change from 2018 (ha) | Area (ha) | Change from 2018 (ha) | Area (ha) | Change from 2018 (ha) |
| $x_1$ | Impervious surface | 2136.72 | 2513.63 | 376.91 | 2202.81 | 66.09 | 2401.83 | 265.11 |
| $x_2$ | Forest | 856.36 | 685.71 | 170.65 | 933.43 | 77.07 | 793.44 | 62.92 |
| $x_3$ | Cropland | 4601.00 | 4574.88 | 26.12 | 4536.90 | 64.10 | 4561.83 | 39.17 |
| $x_4$ | Grassland | 87.66 | 88.56 | 0.90 | 84.33 | 3.33 | 74.79 | 12.87 |
| $x_5$ | Shrubland | 173.26 | 224.36 | 51.10 | 216.58 | 43.32 | 237.61 | 64.35 |
| $x_6$ | Water | 605.84 | 598.23 | 7.61 | 624.24 | 18.40 | 608.22 | 2.38 |
| $x_7$ | Bare land | 309.85 | 85.32 | 224.53 | 171.36 | 138.49 | 92.97 | 216.88 |
| $f_1(x)$ | Economic benefit ($10^4$ CNY) | $3.47 \times 10^6$ | $4.06 \times 10^6$ | $5.90 \times 10^5$ | $3.50 \times 10^6$ | $1.15 \times 10^5$ | $3.89 \times 10^6$ | $4.2 \times 10^5$ |
| $f_2(x)$ | Ecological benefit ($10^4$ CNY) | $7.23 \times 10^3$ | $6.91 \times 10^3$ | $-321.58$ | $7.46 \times 10^3$ | 233.17 | $7.16 \times 10^3$ | $-71.56$ |

### 4.2.2. Ecological Priority Scenario

The ecological benefits in the study area significantly increased to $7.46 \times 10^{11}$ CNY, an increase of $2.33 \times 10^7$ CNY compared with 2018. However, economic benefits only increased by 3.2%. In this scenario, the impervious surface in the study area increased by 66.09 ha compared with 2018, indicating continued growth in construction land to meet the trend of social development. Cropland, grassland, and bare land decreased in comparison with 2018, with reductions of 64.1 ha, 3.33 ha, and 138.49 ha, respectively. Forest, shrubland, and water bodies increased by 78.11 ha, 43.32 ha, and 18.4 ha, respectively. The urban expansion process resulted in a decrease in cropland, grassland, and bare land, but due to the maximization of ecological benefits, the scale of urban expansion was limited, and the increase in economic benefits could not be guaranteed.

### 4.2.3. Balanced Development Scenario

To comply with the laws of social development and ensure sustainable economic growth while maintaining resource and environmental sustainability, the study area achieved economic benefits of $3.89 \times 10^{10}$ CNY and ecological benefits of $7.16 \times 10^7$ CNY. Under this scenario, the expansion scale of construction was somewhat limited. The area of construction land increased by 265.11 ha, 199.02 ha more than the Ecol. Prior. scenario and 111.80 ha less than the Econ. Prior. scenario. The area of shrubland increased by 64.35 ha, while forest, cropland, grassland, and bare land experienced varying degrees of reduction, and the water remained relatively stable.

### 4.3. Spatial Layout Optimization Results

According to the model accuracy validation in Section 4.1, it was known that the PLUS model was feasible for simulating land use changes at the township level based on the validation of pixel ratio, Kappa, and the FoM value. The spatial layout optimization results in 2035 were shown in Figure 3. To analyze the urban expansion patterns under different development scenarios, the area of impervious surface and its comparison with the 2018 status were statistically summarized by administrative district, as demonstrated in Table 7.

**Table 7.** Statistics and changes in the impervious surface area of various administrative districts.

| Administrative Division | Impervious Surface Area in 2018 (ha) | Different Development Scenarios in 2035 | | | | | |
|---|---|---|---|---|---|---|---|
| | | Econ. Prior. Scenario | | Ecol. Prior. Scenario | | BD Scenario | |
| | | Area (ha) | Change from 2018 (ha) | Area (ha) | Change from 2018 (ha) | Area (ha) | Change from 2018 (ha) |
| Shijialiang Town | 217.53 | 264.96 | 47.43 | 289.98 | 72.45 | 277.92 | 60.39 |
| Caijiagang Subdistrict | 1408.14 | 1624.68 | 216.54 | 1428.75 | 20.61 | 1522.53 | 114.39 |
| Tongjiaxi Town | 511.05 | 623.99 | 112.94 | 484.08 | 26.97 | 601.38 | 90.33 |

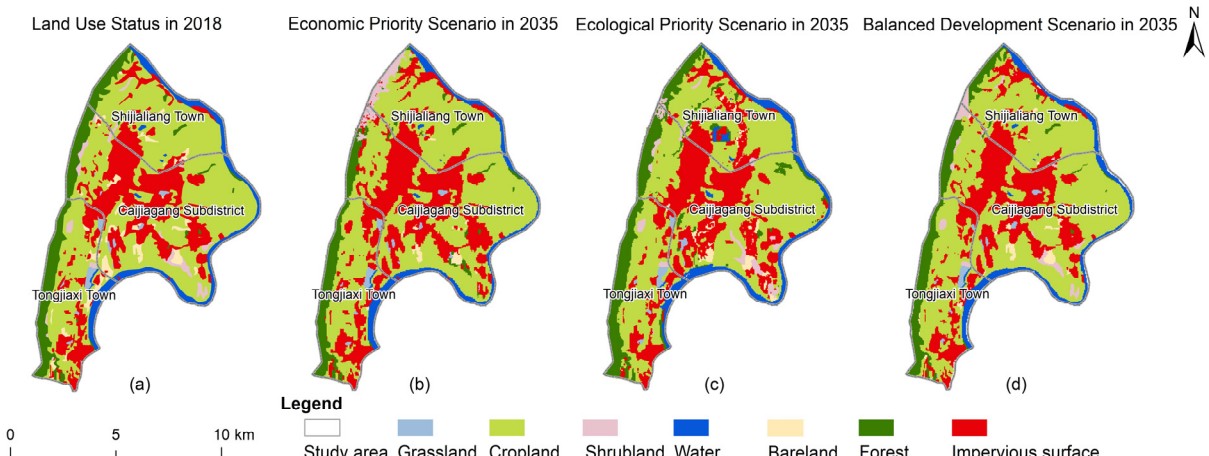

**Figure 3.** (**a**) Land use status in 2018; Multi-scenario spatial layout optimization results in 2035: (**b**) Econ. Prior. scenario; (**c**) Ecol. Prior. scenario; (**d**) BD scenario.

### 4.3.1. Economic Priority Scenario

There was a clear trade-off between forest and impervious surface in the spatial distribution. Construction land in the three administrative divisions experienced varying degrees of expansion, while bare land, shrubland, and forest showed reductions. Compared with 2018, the construction land increased by 47.43 ha in Shijialiang Town, 216.54 ha in Caijiagang Subdistrict, and 112.94 ha in Tongjiaxi Town. Shijialiang Town and Tongjiaxi Town continued expanding around their existing construction land, concentrating urban growth through encroachments on forest, bare land, and shrubland, following the general pattern of urban development. Shijialiang Town mainly increased construction land by encroaching on forest, resulting in scattered expansion. Limited by the influence of slope, urban expansion was mainly in relatively flat areas. The increased construction land in Tongjiaxi Town was primarily centralized in the western and southern regions, where original bare land was developed to create new urban growth areas. Caijiagang Subdistrict had a relatively smaller extent of building land expansion, but it had the largest area of construction land. It expanded from the western edge of the study area along the edge of the forest towards the eastern direction, showing a tendency to approach the original city center. The west of the study area was relatively higher in elevation, and urbanization was achieved by encroaching on surrounding bare land and shrubland based on the existing construction land. The northwest of the study area displayed a scattered and uncoordinated distribution. This layout does not support the promotion of integrated urban-rural development, rational infrastructure construction, or long-term urban development.

### 4.3.2. Ecological Priority Scenario

The spatial layout of land use differed from the Econ. Prior. scenario. In all three regions, the area of impervious surfaces decreased to varying degrees. Simultaneously, the forest area exhibited significant growth, expanding upon the sparse forest base. Strip-shaped forest regions appeared at the urban edges. Some scattered and fragmented rural

construction land gradually degraded into cropland, and the existing residents gathered in larger neighboring towns. Compared with 2018, the impervious surface area increased by 72.45 ha in Shijialiang Town and 20.61 ha in Caijiagang Subdistrict. Shijialiang Town continued to develop based on its original construction land, with significant growth in the central and southern parts. Caijiagang Subdistrict, located at the center of the Caijia Cluster, took advantage of its relatively flat terrain. By 2018, urbanization had reached a considerable magnitude. In 2035, some scattered construction land was reclamation, ensuring the rational utilization of land and conforming to the concept of integrated urban–rural development. Therefore, the expansion extent of construction land was not prominent. The area in Tongjiaxi Town experienced a decrease of 26.97 ha due to the integration of some small villages in the western region, forming a concentrated area. Under the scenario of ecological protection, the urbanization scale obviously cut down.

### 4.3.3. Balanced Development Scenario

The expansion of construction land in the three districts continued outward spatially. Forest, shrubland, and other land types underwent significant changes due to the combined effects of multiple driving factors. Compared with 2018, the impervious surface area in Shijialiang Town, Caijiagang Subdistrict, and Tongjiaxi Town increased by 60.39 ha, 114.39 ha, and 90.33 ha, respectively. Among them, Shijialiang Town has a relatively larger urban scale, with noticeable expansion in the west. Based on original construction land, its urbanization was accomplished by expanding into surrounding cropland, shrubland, and barren land, capitalizing on the proximity to water resources. Caijiagang Subdistrict experienced a slight westward expansion of construction land. Under the BD scenario, it ensured not only the growth of construction land with high economic benefits but also a relatively stable area for shrubland and forest, thereby guaranteeing certain ecological benefits. The distribution of construction land in Tongjiaxi Town in 2018 was relatively fragmented. Its urbanization mainly depended on the construction of surrounding bare land, utilizing the existing urban foundation and the advantage of water resources to achieve eastward expansion and gradually form a relatively concentrated belt-shaped center.

## 5. Discussion

We innovatively applied the PLUS model for simulating and predicting land use changes at the township scale. Multi-scenarios were set using the MOP model, and the PLUS model was used to simulate and compare land use spatial differences. Various factors were considered in this study, including land use planning policies (such as changes in the area of construction land, forest, etc.), population, and ecological environmental dynamics. The constraints for the MOP model were established. Our study was to create a clear and easy-to-understand optimization model that took into account regional land use demands and environmental pressures. It helped determine how the quantity of land use changed under different development scenarios in the region. Under the Econ. Prior. scenario, the construction land expanded rapidly, the incremental economic benefits of the city peaked, and the ecological benefits were severely regressed. Under the Ecol. Prior. scenario, urban expansion was realized based on guaranteeing the greening rate, so the increase in the city's economic benefits was not large, while the ecological benefits showed a significant increase. Under the BD scenario, the land use structure change aligned with the future development trends; avoided the blind encroachment of other non-construction land in the process of urbanization; and compensated for the insufficient growth of economic benefits under ecological protection. In all three scenarios, there was an expansion of construction land. It complied with the constraints for a future land use development rate, which must not be lower than the rate in 2018. Additionally, the scenarios were in accordance with urbanization trends. Changes in the areas of ecological lands, such as forests and shrubland, also conformed to planning policies. This study simulated three scenarios based on policy guidance. It is important to note that these scenarios cannot represent all possible land use situations. Nevertheless, we firmly believe that regional development should strive

to explore a balance point between ecological and economic development. Therefore, we proposed the BD scenario as the direction for regional urban expansion. It contributed significantly to the long-term sustainability of the region, and it was consistent with the previous findings [51,52].

The accuracy validation method of the PLUS model is concise and straightforward. The selection and setting of driving factors and parameters are two main aspects [53] that influence the model's simulation accuracy. Given the accessibility and effectiveness of data and the specific characteristics of the study area, this study utilized a sufficient number of constraint conditions and driving factors to optimize and simulate land use changes. The results adhered to the principles of socio-economic development. In addition, we considered the changes in the accuracy of land use patches in both quantitative and spatial aspects, verified the quantitative reasonableness with the correct proportion of image elements for land use change simulation, and verified the spatial reasonableness with the Kappa and the FoM indices. Regarding the PLUS model's accuracy in small-scale mountainous areas, the results had higher simulation accuracy compared with the FLUS model (Kappa increased from 0.67 to 0.73 and FoM increased from 0.1962 to 0.263) [32]; it can be assumed that the simulation results of the PLUS model will be more reliable under different scenarios in the future [44]. Therefore, the MOP-PLUS model is feasible for the multi-scenario simulation of land use change at the scale of mountain townships.

We explored the applicability of the MOP-PLUS model for simulating land use changes in small-scale mountainous areas. Positive results were achieved, offering innovative insights into the methodology for land use planning and urban development goal setting in small-scale areas. Furthermore, our study, which focused on land use changes within a specific geographical context and scale, has expanded the scope of land use change simulation research. The simulated results of land use changes aligned with the general processes of urbanization and can support the theoretical development of the simulation and optimization of land use change within urbanization processes. In the face of the unique challenges posed by mountainous urban terrain and ecological conservation, our research findings contribute to aiding regional land use policy formulation and providing theoretical guidance for land use management. This facilitates the intersection of land use theory and practice.

## 6. Conclusions

This study tried to construct the MOP-PLUS method to realize land use optimization and explore the applicability of the PLUS model in small-scale mountainous areas. The conclusions are as follows:

(1) The MOP-PLUS model has shown the feasibility in simulating land use change patterns in small-scale mountainous areas. The study integrated the MOP and the PLUS models using a "quantity + spatial" modeling approach, which considered macro policies, regional area, population, and spatial characteristics such as topography and transportation. By combining the strengths of both models and addressing their weaknesses, we aimed to achieve the optimal land use pattern that promotes urban development. The OA of land use simulation reached 81.6%, indicating the high correctness of the PLUS model in small-scale areas. The Kappa value was 0.76 in this study, suggesting that the PLUS model's spatial layout optimization has a reasonable effect. The simulation results of multi-scenarios have shown the feasibility of the PLUS model in simulating land use changes at a smaller scale, helping to improve the land use spatial layout optimization efficiency.

(2) The BD scenario demonstrated relatively optimal urban development benefits and spatial layout. (1) In the Econ. Prior. scenario, the economic and ecological benefits were $4.06 \times 10^{10}$ CNY and $6.91 \times 10^7$ CNY. The area of impervious surface increased the most by 17.64%, characterized by an uncontrolled growth of construction land and an increased fragmentation of patches. Ecological protection was largely disregarded in this urban expansion pattern, which is unfavorable to sustainable urban development. (2) In the Ecol. Prior. scenario, the economic and ecological benefits were $3.50 \times 10^{10}$ CNY and

$7.46 \times 10^7$ CNY. The forest expanded by 9.00%, while the area of the impervious surface merely increased by 3.09%. The construction land exhibited a more concentrated layout within town centers, where it was dispersed at the edges. The presence of patch fragmentation indicated urban regression. (3) In the BD scenario, the economic and ecological benefits were $3.89 \times 10^{10}$ CNY and $7.16 \times 10^7$ CNY. The impervious surface area increased by 12.41%, while the forest decreased by 7.34%. The construction land tended to concentrate spatially in this scenario. It avoided uncontrolled expansion and maintained a dynamic balance among construction land and ecological land.

(3) The BD scenario was the relatively optimal urban development strategy. Among the three scenarios simulated by the MOP-PLUS model, the development mode prioritizing construction land expansion resulted in the highest urban development benefits. However, it led to significant ecological loss, highlighting a notable drawback of urbanization focused on economic development. Meanwhile, a single ecological benefit objective presents similar issues. Based on a comprehensive consideration of the regional land use constraint policies, we believe that a balance point for land use demands can be found in the BD scenario. It can ensure economic growth without compromising the ecological environment. And it promotes construction land-concentrated development and efficient resource utilization, benefiting various aspects such as healthcare and education, providing decision-makers with valuable insights for planning future development directions and better serving the cause of sustainability.

**Author Contributions:** Methodology, validation, and formal analysis, Yuqing Zhong and Yanfei Yang; resources and data curation, Yuqing Zhong and Minghui Xue; writing—original draft preparation, Yuqing Zhong; writing—review and editing, Xiaoxiang Zhang; visualization, Yuqing Zhong; supervision, Xiaoxiang Zhang. All authors have read and agreed to the published version of the manuscript.

**Funding:** This article was written as part of the work of the Centre for Sustainable, Healthy and Learning Cities and Neighbourhoods (SHLC), which is funded via UK Research and Innovation, and administered through the Economic and Social Research Council, as part of the UK Government's Global Challenges Research Fund. Project Reference: ES/P011020/1.

**Data Availability Statement:** Not applicable.

**Conflicts of Interest:** The authors declare no conflict of interest.

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
