# Peer review of "Optimization and Simulation of Mountain City Land Use Based on MOP-PLUS Model: A Case Study of Caijia Cluster, Chongqing"

_ijgi, doi:10.3390/ijgi12110451_

Round 1
Reviewer 1 Report (Previous Reviewer 1)
Comments and Suggestions for Authors
After a round of revisions, the article is generally acceptable.
Author Response
Dear Reviewer,
Thank you for your feedback and for taking the time to review our article. We appreciate your efforts in helping us improve the quality of our work.
We are pleased to hear that after our revisions, the article is generally acceptable. Your guidance and suggestions have been invaluable in making the necessary improvements. If there are any specific areas or aspects you believe still require further attention or refinement, please do not hesitate to let us know, and we will address them promptly.
Once again, we sincerely appreciate your valuable input, and we look forward to your continued support and guidance as we work to enhance the article further.
Best regards,
Yuqing Zhong
Reviewer 2 Report (Previous Reviewer 2)
Comments and Suggestions for Authors
The author's revision and response are very detailed. However, it is suggested to continue to improve. Specific as follows:
1. The picture quality of the paper should be further improved, which is not clear and standardized enough.
2. The conclusion of the paper should be further streamlined and focused, and the focus is not prominent at present.
3. The discussion part needs to summarize the policy implications, which corresponds to the research process and results. You can consider hierarchical description.
4. In the overall structure, the order of discussion and conclusion can be reversed.
Comments on the Quality of English LanguageModerate editing of English language required.
Author Response
Dear Reviewer,
Thank you for your valuable feedback on our manuscript. In response to your comments, we have provided a detailed response in the attachment.
Please see the attachment.
Once again, we sincerely thank you for your time and expertise in reviewing our work.
Best regards,
Yuqing Zhong

Reviewer 3 Report (Previous Reviewer 4)
Comments and Suggestions for Authors
The reviewer sent the editorial office his/her comments.
Author Response
Dear Reviewer,
Thank you for your careful review and helpful comments. We appreciate your efforts in helping us improve the quality of our work.
Your feedback and comments are valuable to us in making the necessary improvements. If there are any specific areas or aspects that you feel still need further attention or improvement, please feel free to let us know and we will address them in a timely manner.
Once again, we sincerely appreciate your helpful comments and look forward to your continued support and guidance in further improving this article.
Best regards,
Yuqing Zhong
This manuscript is a resubmission of an earlier submission. The following is a list of the peer review reports and author responses from that submission.
Round 1
Reviewer 1 Report
Comments and Suggestions for Authors
Report on the manuscript IJGI-2556134
This article focus on the land use of the town areas typically in the mountain region. Given on the PLUS model, and based on the economic priority, ecological priority, and balanced development scenarios, evidences show that PLUS model is useful for the small region analysis which can effectively increase the land use.
Major issue:
1. Why set so many restrictions? Is that means your pre-analysis reflects some disequilibrium consequences?
2. Results in Table 6 show the economic and ecological benefits under 3 scenarios. Is it correct that the ecological benefit under the balanced development scenario (71.56) is smaller than the ecological benefit under the ecological scenario (2.33*10^2)?
3. My understanding is that, you use scenarios to simulate the future situation of the STUDY AREA (Fig. 1), and derive Fig. 3. How to model and scenarios dealing with the situations of the boundary regions? Is it more suitable to simulate a relative larger area but focus on the study area?
4. A major issue of this study is that, you want to show that PLUS has the ability to solve some issues, but there is no comparisons with other models. If accuracies of other models are significantly higher than PLUS model, then at least PLUS is not quite acceptable for the land use design of the mountain type town areas.
Minor issue:
1. In Table 3, please provide the sources of the population coefficients, the sources of the elasticity coefficient, and sources of all other unclear values.
2. Please explain more about equation (10) and (11). I am a bit lost myself here.
3. Table 4, suggest adjusting the table width.
4. Equation (12) and (13), italic.

Reviewer 2 Report
Comments and Suggestions for Authors
1. Introduction
In the last paragraph of the introduction, the author needs to better describe the innovative points and scientific issues to be addressed in this article.
2. Material and data preprocessing
The description in this chapter is not clear enough, and it needs to be explained better. What is the specific date of the Landsat image used.
3. Method
The description of the method should be more concise, reviewing the entire text and attempting to separate the results from the method.
5. Conclusion
The accuracy analysis of the method needs to be added in the conclusion section.
6. Discussion
The discussion section should provide a more in-depth description, which is currently insufficient.
1) Further discussion is needed on the effectiveness and quantitative results of the proposed method.
2) It is best to see the advantages and improvements of this study compared to other studies.
Reviewer 3 Report
Comments and Suggestions for Authors
-
1.The study sample is too small.
-
2.The article fails to further theorize and rise to a general and regular academic theory development
Extensive editing of English language required
Reviewer 4 Report
Comments and Suggestions for Authors
The reviewer sent the comments to the editorial office.
Comments on the Quality of English Language-